# Sarcopenia increases the risk of post-operative recurrence in patients with non-small cell lung cancer

**Yo Kawaguchi**[ORCID]*, **Jun Hanaoka**[◉], **Yasuhiko Ohshio**[◉], **Keigo Okamoto**[◉], **Ryosuke Kaku**[◉], **Kazuki Hayashi**[◉], **Takuya Shiratori**[◉], **Akira Akazawa**[◉]

Division of General Thoracic Surgery, Department of Surgery, Shiga University of Medical Science, Shiga, Japan

◉ These authors contributed equally to this work.
* kawaguchi1228@yahoo.co.jp

**Data Availability Statement:** All relevant data are within the manuscript and its Supporting information files.

## Abstract

### Background

Sarcopenia is among the most prevalent and serious cancer-related symptom, and is strongly correlated with a poor prognosis. Moreover, it reportedly predicts poor prognosis after surgery in patients with lung cancer. However, it is unclear whether sarcopenia directly affects post-operative recurrence. The purpose of this study was to evaluate whether sarcopenia can be a risk indicator for post-operative recurrence, and whether it suppresses anti-tumor immunity, in a cohort of patients with resected non-small cell lung cancer.

### Methods

This study retrospectively reviewed the data of 256 consecutive patients who underwent curative lobectomy and lymph node dissection for non-small cell lung cancer at our institution. The psoas muscle mass index was calculated as the total psoas muscle area at the third lumbar vertebral level/height$^2$ (cm$^2$/m$^2$). Sarcopenia was defined by a psoas muscle mass index of under 5.03 cm$^2$/m$^2$ and 3.17 cm$^2$/m$^2$ in male and female patients, respectively. Post-operative prognosis and cumulative incidence of recurrence rates were calculated.

### Results

The 5-year overall survival and disease-free survival rates post-surgery were 59.5% and 38.6%, respectively, in patients with sarcopenia versus 81.1% and 72.1%, respectively, in patients without sarcopenia ($p < 0.001$). The 5-year cumulative incidence of recurrence rate in patients with sarcopenia was significantly higher than those without sarcopenia (49.9% versus 22.4%, respectively) in every pathological stage. Pathological stages II and III (hazard ratio, 3.36; $p = 0.004$), histological type (hazard ratio, 2.31; $p = 0.025$), and sarcopenia (hazard ratio, 2.52; $p = 0.001$) were independent risk factors for post-operative recurrence according to multivariate analysis.

**Funding:** The authors received no specific funding for this work.

**Competing interests:** The authors have declared that no competing interests exist.

## Conclusion

Sarcopenia is a risk indicator for post-operative recurrence in patients with non-small cell lung cancer.

## Introduction

Non-small cell lung cancer (NSCLC) is a major cause of cancer-related deaths globally [1]. Although tumor characteristics, such as histology and TNM stage, and treatment factors, such as surgery and chemotherapy, have traditionally been considered important to determine cancer prognosis, recently, patient factors like sarcopenia have started gaining attention [2]. Significant loss of muscle (sarcopenia/cachexia) is among the most prevalent and serious cancer-related symptoms and is strongly correlated to poor prognosis [2], especially in patients with advanced cancer. Several studies have recently reported that sarcopenia in patients with resectable cancers including gastric cancer [3], colorectal cancer [4], pancreatic cancer [5], hepatocellular carcinoma [6], endometrial cancer [7], renal cell carcinoma [8], and lung cancer [9] also contributes to poor post-operative survival. Poor prognosis in sarcopenic patients after surgery may be attributed to two reasons. First, sarcopenia itself is generally accepted to confer poor prognosis. Even if the cancer is completely cured through surgery, there is loss of skeletal muscle mass and strength, resulting in physical disability and death [10]. Second, sarcopenia may also contribute to the risk of cancer recurrence [10].

Skeletal muscle has recently been identified as a secretory organ, producing myokines; these may activate immune cells to suppress the growth of cancer cells [11]. Although some reports have demonstrated that sarcopenia decreases post-operative overall survival (OS) [9,12–19] and disease-free survival (DFS) [9,14,18] in NSCLC patients, it is currently unclear whether sarcopenia directly affects post-operative recurrence.

Therefore, the purpose of this study was to demonstrate whether sarcopenia can be a risk indicator for post-operative recurrence, and to examine whether sarcopenia suppresses anti-tumor immunity, in a cohort of patients with resected NSCLC.

## Methods

### Patients

This study was approved by the institutional review board of the Shiga University of Medical Science (Approval Number: R2020-048). The need for informed consent was waived by the institutional review board owing to the retrospective nature of the study. We accessed patients' medical records between April, 2020 and June 2021. These records were all anonymized during data collection; we used the anonymized data for our analysis.

This study reviewed the data of 256 consecutive patients who underwent curative lobectomy and lymph node dissection for NSCLC at our institution between January 2011 and December 2015. We collected data on the following patient characteristics: age, sex, performance status, smoking history, adjuvant chemotherapy, maximum tumor size, short diameter of lymph nodes which was maximum in mediastinum or hilar lymph nodes, pathological stage, histological type, lymphocyte and monocyte pre-operative blood examination values, and the cross-sectional area (cm$^2$) of the psoas muscle at the third lumbar vertebral (L3) level on computed tomography (CT) scans. Histological assessment was performed according to the histological classification of the World Health Organization. The seventh edition of the

TNM classification for lung cancer was used for pathological staging. Hematological data was collected from each patient within two weeks prior to surgical treatment. We also recorded the pre-operative lymphocyte-to-monocyte ratio (LMR), which was calculated by dividing the absolute lymphocyte counts by the absolute monocyte counts. We used an LMR cut-off value of 3.2, as previously reported [20,21]. We excluded the following cases: limited resection, incomplete resection, and no CT image at the L3 level.

## Definition of sarcopenia

We used the psoas muscle mass index (PMI), which had been reported by Hamaguchi et al. to be a diagnostic tool for sarcopenia [22] and was often used in many reports to predict post-operative outcomes in patients with NSCLC [9,16–18]. Preoperative CT scans were performed within three months prior to surgery. The cross-sectional areas (cm2) of the right and left psoas muscles at the L3 level were measured by manual tracing of CT data using Shade Quest View R (Fujifilm, Japan). Subsequently, the total psoas area was normalized for height. The PMI was calculated as previously described [22]: PMI = total psoas muscle area at the L3 level/ height$^2$ (cm$^2$/m$^2$). Since no consensus currently exists on the PMI cut-off value of sarcopenia to predict recurrence, we defined sarcopenia as a PMI of under 5.03 cm$^2$/m$^2$ in male patients and 3.17 cm$^2$/m$^2$ in female patients. We derived these cut-off values to be good indicators for recurrence. In predicting recurrence, this definition had a sensitivity of 67.2%, specificity of 57.1%, and area under the receiver operating characteristic curve value of 0.631 in males; in comparison, those for females were 68.4%, 59.4%, and, 0.590, respectively (S1 Fig). In addition, we created another group which included patients who underwent curative lobectomy and lymph node dissection for NSCLC at our institution in 2016; this was the validation cohort. In this cohort, we analyzed the association between sarcopenia and post-operative recurrence using our PMI cut off values, and assessed the plausibility.

## Post-operative outcomes

Post-operative outcomes were represented by the OS, DFS, and cumulative incidence of recurrence (CIR). OS was defined as the interval between the date of surgery and the date of death. DFS was defined as the interval between the date of surgery and the date of death or cancer recurrence (confirmed using an imaging test). CIR was defined as the interval between the date of surgery and the date of cancer recurrence (confirmed using an imaging test) and was set as the primary variable. Most patients were followed up postoperatively every three or six months, until the fifth year after surgery. Screening examination included tumor markers, chest and abdominal CT, and cerebellar magnetic resonance imaging. Bone scintigraphy and positron emission tomography CT were performed on demand. The 5-year OS, DFS, and CIR were compared between patients with and without sarcopenia.

## Statistical analyses

Statistical analyses were performed using SPSS Statistics for Windows, version 25 (SPSS Inc., Chicago, IL, USA). Associations between variables were analyzed using Fisher's exact test. The t-test was also used to compare the means of parametric data between the two groups. The Kaplan-Meier method was used to determine OS, DFS, and CIR. The log-rank test was used to compare survival differences for each variable. The Cox's proportional hazards model was used for multivariate analysis. Statistically significant differences were defined as $p < 0.05$.

## Results

### Patient characteristics

During the study period, 354 patients received surgery for NSCLC at our institution. Among them, we excluded cases with limited and incomplete resections; 256 cases of curative lobectomy and lymph node dissection were therefore included in this study. The median follow-up period was 44.9 months. The patient characteristics are shown in Table 1. The mean age of patients with sarcopenia were higher than those without, and there were no significant differences in the distribution of sarcopenia patients among various pathological stages. Adenocarcinomas were fewer and LMRs of ≦3.2 were more common in sarcopenia patients.

### Survival and recurrence comparison between sarcopenia and non-sarcopenia patients

The 5-year OS and DFS after surgery were 59.5% and 38.6%, respectively, in patients with sarcopenia versus 81.1% and 72.1%, respectively, in patients without sarcopenia ($p < 0.001$) (Fig 1). Since the direct influence of sarcopenia on cancer recurrence could not be estimated using OS and DFS, we examined CIR. The 5-year CIR rate in patients with sarcopenia was 49.9%, which was significantly higher than the 22.4% observed in patients without sarcopenia (Fig 2). Patients with and without sarcopenia were further divided according to pathological stage. The 5-year CIR rate in patients with sarcopenia was higher in every stage than those without sarcopenia (stage I: 29.8% in sarcopenia versus 5.5% in non-sarcopenia; $p < 0.001$, stage II: 70.0% in sarcopenia versus 40.0% in non-sarcopenia; $p = 0.030$, stage III: 86.7% in sarcopenia versus 56.1% in non-sarcopenia; $p < 0.087$) (Fig 3).

To assess the plausibility of our original PMI cut off values, we analyzed prognosis using the validation cohort, which included 36 sarcopenia and 46 non-sarcopenia patients. The 5-year OS and DFS after surgery were 77.9% and 50.2%, respectively, in patients with sarcopenia versus 88.0% and 66.5%, respectively, in patients without sarcopenia (S2 Fig). The 5-year CIR rate was 48.0% in patients with sarcopenia versus 29.3% in patients without sarcopenia (S3 Fig). Based on these results, we considered our PMI cut off values to be plausible, as we could demonstrate the association between sarcopenia and post-operative recurrence in the validation cohort.

### Association between recurrence and clinicopathological factors

We identified the following potential risk factors for recurrence using the comparison between sarcopenia and other factors commonly recognized to predict recurrence; these included: age over 65 years, male sex, performance status 1–2, smoking history, absence of adjuvant chemotherapy, large tumor size, large lymph node size, pathological stage II and III, histological type (other than adenocarcinoma and squamous cell carcinoma), grade of differentiation, lymphatic permeation, vascular invasion, sarcopenia, and low LMR (Table 2). According to multivariate analysis, pathological stage II and III (hazard ratio, 3.36; $p = 0.004$), histological type (hazard ratio, 2.31; $p = 0.025$), and sarcopenia (hazard ratio, 2.52; $p = 0.001$) were independent risk factors for post-operative recurrence (Table 3).

## Discussion

In our cohort, sarcopenia was the risk indicator for post-operative recurrence in every stage. Reports have previously demonstrated that sarcopenia decreased OS [9,12–19] and DFS [9,14,18] after surgery in patients with NSCLC. In contrast to these previous studies, our study has two unique features. First, using CIR, we are probably the first to demonstrate that

**Table 1. Characteristics of patients with and without sarcopenia.**

| Variables | Sarcopenia | | Non-Sarcopenia | | p value |
|---|---|---|---|---|---|
| | N = 128 | % | N = 128 | % | |
| **Age, years** | | | | | 0.002 |
| Median | 70.1 | - | 66.9 | - | |
| Range | 51–88 | | 47–83 | | |
| **Sex** | | | | | 0.504 |
| Male | 89 | 69.5 | 84 | 65.6 | |
| Female | 39 | 30.5 | 44 | 34.4 | |
| **Performance status** | | | | | 0.062 |
| 0 | 96 | 75.0 | 108 | 84.4 | |
| 1–2 | 32 | 25.0 | 20 | 15.6 | |
| **Smoking history** | | | | | 0.070 |
| (+) | 100 | 78.7 | 88 | 68.8 | |
| (-) | 27 | 21.3 | 40 | 31.2 | |
| **Psoas mass index (Male)** | | | | | <0.001 |
| Median | 4.05 | - | 6.09 | - | |
| Range | 1.97–5.02 | | 5.03–8.58 | | |
| **Psoas mass index (Female)** | | | | | <0.001 |
| Median | 2.67 | | 4.15 | | |
| Range | 2.00–3.16 | | 3.17–5.48 | | |
| **LMR** | | | | | 0.037 |
| ≤ 3.2 | 36 | 28.1 | 23 | 18.0 | |
| > 3.2 | 92 | 71.9 | 105 | 82.0 | |
| **Adjuvant chemotherapy (stage IB–III)** | | | | | 0.015 |
| (+) | 44 | 55.0 | 54 | 75.0 | |
| (-) | 36 | 45.0 | 18 | 25.0 | |
| **Tumor size (mm)** | | | | | 0.002 |
| Median | 30.9 | - | 23.8 | - | |
| **Lymph node size** | | | | | 0.002 |
| >1cm | 29 | 22.7 | 11 | 8.6 | |
| ≦1cm | 99 | 77.3 | 117 | 91.4 | |
| **Pathological stage** | | | | | |
| I | 70 | 54.7 | 76 | 59.4 | 0.449 |
| II | 34 | 26.6 | 28 | 21.9 | 0.381 |
| III | 24 | 18.8 | 24 | 18.8 | 1.000 |
| **Histological type** | | | | | |
| Adenocarcinoma | 78 | 60.9 | 95 | 74.2 | 0.023 |
| Squamous cell carcinoma | 35 | 27.3 | 24 | 18.8 | 0.103 |
| Other | 15 | 11.7 | 9 | 7.0 | 0.198 |
| **Grade of differentiation** | | | | | 0.125 |
| 1 | 47 | 38.2 | 53 | 48.2 | |
| 2–4 | 76 | 61.8 | 57 | 51.8 | |
| **Lymphatic permeation** | | | | | 0.401 |
| (+) | 71 | 59.2 | 64 | 53.8 | |
| (-) | 49 | 40.8 | 55 | 46.2 | |
| **Vascular invasion** | | | | | 0.019 |
| (+) | 91 | 75.8 | 70 | 58.8 | |
| (-) | 29 | 24.2 | 49 | 41.2 | |

LMR, lymphocyte-to-monocyte ratio.

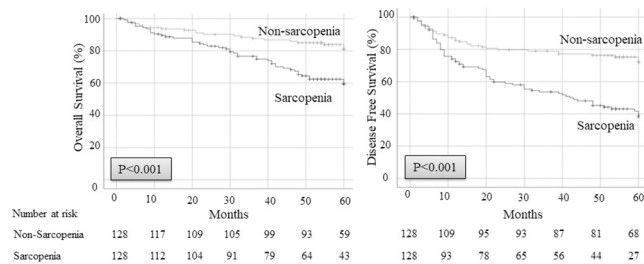

**Fig 1. OS and DFS after surgery compared between sarcopenia patients and non-sarcopenia patients.** DFS, disease-free survival; OS, overall survival.

sarcopenia in patients with lung cancer directly affected post-operative recurrence. We could not determine whether sarcopenia directly affected cancer recurrence using OS or DFS, because decreases in OS or DFS are not only attributable to lung cancer death/recurrence, but also to deaths from other diseases. Indeed, sarcopenia itself is a poor prognostic factor, because the loss of muscle mass and strength results in physical disability and death [10,23]. Second, we focused on lobectomy as the surgical procedure. Previous studies showed heterogeneity in the surgical procedures included, such as partial resection, segmentectomy, and pneumonectomy; this could have affected the recurrence rate and prognosis.

We found sarcopenia to be an independent risk indicator for recurrence. Using multivariate analysis, we compared the risk of recurrence using several factors that are commonly recognized to predict recurrence. We found that advanced pathological stage, histological type (other than adenocarcinoma and squamous cell carcinoma), and sarcopenia were independent risk factors for post-operative recurrence. Interestingly, the development of sarcopenia causes a higher risk of recurrence than that of many tumor factors including histological type, grade of differentiation, lymphatic permeation, and vascular invasion (Table 3).

There may be two reasons for the association between sarcopenia and post-operative recurrence. First, tumors in sarcopenia patients have higher malignant potential than those in

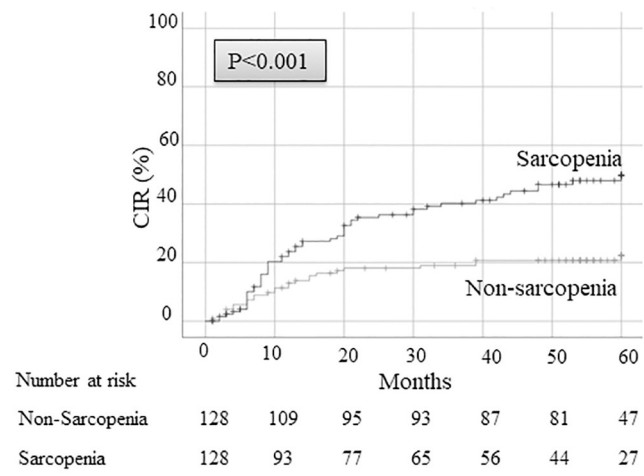

**Fig 2. CIR rate after surgery between sarcopenia patients and non-sarcopenia patients.** CIR, cumulative incidence of recurrence.

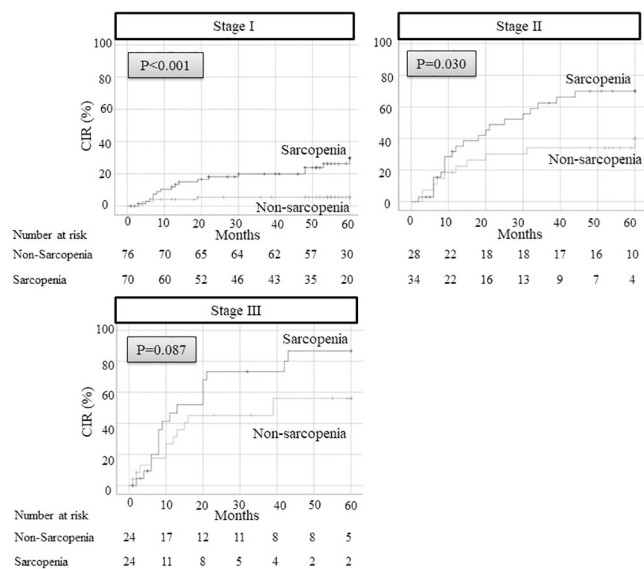

**Fig 3. CIR rate after surgery between sarcopenia patients and non-sarcopenia patients divided according to their pathological stages.** CIR, cumulative incidence of recurrence.

patients without sarcopenia. As a result, even if the pathological stages are equivalent, tumor recurrence occurs more frequently in the former [13]. In the present study, NSCLC patients with sarcopenia presented with larger tumor size, larger lymph node size, and more vascular invasion than those without sarcopenia. Second, muscles exert a potential tumor inhibitory effect via myokines; skeletal muscle has recently been identified as a secretary organ, and is reported to activate immune cells [11]. To examine the association between sarcopenia and pro-tumor/anti-tumor immunity, we used LMR, which can reflect the tumor suppressing activity of tumor-infiltrating lymphocytes and the pro-tumor activity of macrophages in the tumor microenvironment [20]. Therefore, an elevated LMR is favorable for anti-tumor immunity, while a decrease favors pro-tumor immunity. In our study, post-operative recurrences were significantly more common in patients with low LMR, and sarcopenia patients frequently exhibited a low LMR. Previous studies have reported an association between decreased lymphocytes and sarcopenia. For example, a myokine, namely, interleukin (IL)-15 was reported to have an important role in T cell proliferation [24]; exercise induces myokine IL-6, which releases and activates natural killer cells, thereby reducing tumor growth [25]. Therefore, active lymphocytes may be decreased in patients with sarcopenia, predisposing them to recurrences. One study has reported an association between increasing monocytes and sarcopenia, where the myokine irisin dampened macrophage activity in vitro [26]. Therefore, monocytes may increase in patients with sarcopenia due to the decrease in irisin, leading to recurrence. It has been reported that the peripheral monocyte count is associated with the density of the tumor-associated macrophages; this is in turn associated with tumor progression and metastasis [27].

It is unclear whether therapeutic intervention for sarcopenia can prevent post-operative recurrence. However, we believe that sarcopenia could be a treatment target to improve the outcome of NSCLC patients who have undergone surgery. One of the therapeutic approaches for sarcopenia is exercise. Chen et al. demonstrated that patients who engaged in exercise regularly during the first six months after diagnosis have significantly higher OS and DFS (hazard ratios: 0.62 and 0.39, respectively) [28]. Thus, cancer patients should not only exercise because

**Table 2. Association between CIR rate and clinicopathological factors.**

| Variables | N | CIR (%) | p value |
|---|---|---|---|
| **Patient factors** | | | |
| **Age (years)** | | | 0.526 |
| < 65 | 76 | 38.3 | |
| ≧ 65 | 180 | 34.5 | |
| **Gender** | | | 0.048 |
| Male | 173 | 40.4 | |
| Female | 83 | 35.6 | |
| **Performance status** | | | 0.713 |
| 0 | 204 | 35.0 | |
| 1–2 | 52 | 36.9 | |
| **Smoking history** | | | 0.094 |
| (+) | 188 | 38.2 | |
| (-) | 67 | 28.8 | |
| **Sarcopenia** | | | < 0.001 |
| (+) | 128 | 49.9 | |
| (-) | 128 | 22.4 | |
| **LMR** | | | 0.048 |
| ≤ 3.2 | 59 | 44.1 | |
| > 3.2 | 197 | 32.9 | |
| **Treatment factors** | | | |
| **Adjuvant chemotherapy (stage IB-III)** | | | 0.127 |
| (+) | 99 | 53.9 | |
| (-) | 54 | 32.3 | |
| **Tumor factors** | | | |
| **Tumor size** | | | |
| >3cm | 107 | 43.8 | 0.008 |
| >5cm | 40 | 55.2 | < 0.001 |
| **Lymph node size** | | | |
| >1cm | 40 | 52.6 | 0.010 |
| **Pathological stage** | | | |
| I | 146 | 16.7 | < 0.001 |
| II, III | 110 | 61.6 | < 0.001 |
| III | 48 | 70.0 | < 0.001 |
| **Histological type** | | | |
| Adenocarcinoma | 173 | 31.7 | 0.021 |
| Squamous cell carcinoma | 59 | 36.1 | 0.600 |
| Other | 24 | 60.4 | < 0.001 |
| **Grade of differentiation** | | | 0.003 |
| 1 | 100 | 25.6 | |
| 2–4 | 133 | 45.4 | |
| **Lymphatic permeation** | | | < 0.001 |
| (+) | 135 | 50.2 | |
| (-) | 104 | 15.6 | |
| **Vascular invasion** | | | < 0.001 |
| (+) | 161 | 47.3 | |
| (-) | 78 | 11.3 | |

CIR, cumulative incidence of recurrence; LMR, lymphocyte-to-monocyte ratio.

**Table 3. Multivariate analysis on cancer recurrence.**

| Variables | | HR | 95% CI | *p* value |
|---|---|---|---|---|
| **Patient factors** | | | | |
| Age | ≧65 | 1.46 | 0.80–2.65 | 0.214 |
| Gender | Male | 1.00 | 0.45–2.22 | 0.992 |
| Performance status | 1–2 | 0.60 | 0.30–1.23 | 0.166 |
| Smoking history | (+) | 1.11 | 0.47–2.60 | 0.809 |
| Sarcopenia | (+) | 2.52 | 1.48–4.28 | 0.001 |
| LMR | ≦3.2 | 0.97 | 0.85–1.10 | 0.593 |
| **Treatment factor** | | | | |
| Adjuvant chemotherapy | (-) | 1.31 | 0.74–3.13 | 0.252 |
| **Tumor factors** | | | | |
| Tumor size | >5cm | 1.26 | 0.66–2.40 | 0.478 |
| Lymph node size | >1cm | 0.95 | 0.65–1.39 | 0.800 |
| Pathological stage | II, III | 3.36 | 1.47–7.70 | 0.004 |
| Pathological stage | III | 1.70 | 0.88–3.27 | 0.114 |
| Histological type | Other | 2.31 | 1.11–4.82 | 0.025 |
| Grade of differentiation | 2–4 | 0.60 | 0.32–1.14 | 0.122 |
| Lymphatic permeation | (+) | 1.78 | 0.84–3.74 | 0.131 |
| Vascular invasion | (+) | 0.49 | 0.20–1.22 | 0.127 |

LMR, lymphocyte-to-monocyte ratio; HR, hazard ratio; CI, confidence interval.

it improves their overall health, but because as a targeted approach it may improve sarcopenia and decrease the risk of cancer recurrence.

There were several limitations to this study. First, this was a retrospective observational study; some bias associated with the study design or analysis may therefore be inevitable. Second, sarcopenia also occurs in patients suffering from malnutrition, congestive heart failure, chronic obstructive pulmonary disease, chronic renal failure, etc. [23]. Therefore, we cannot exclude the possibility that such potential confounders could have also increased sarcopenia in our patients and influenced cancer recurrence. Third, the definition of sarcopenia varies depending on different studies. Two main diagnostic tools have been proposed for sarcopenia; one used muscle mass only, as reported by Prado et al. in 2008 [29], and one used the presence of low gate speed, decreased grip strength, and low muscle mass, as reported by the European Working Group on Sarcopenia in 2010 [23]. In our study, we used muscle mass to define sarcopenia, because many studies have used this definition in predicting post-operative outcomes in NSCLC patients. Furthermore, we used original cut-off values to define sarcopenia. Therefore, the findings may not be generalized to other cut-off values or cohorts. To address this issue, we analyzed the prognosis using our cut-off values in the validation cohort; this may be considered plausible. Finally, LMR values may vary over time; therefore, we routinely collected the hematological data from each patient within the two weeks before surgical treatment. However, using only one set of hematological data may not offer adequate precision for assessing post-operative prognosis.

## Conclusion

In this study, sarcopenia was a risk indicator for post-operative recurrence in patients with NSCLC. Our findings indicate the need for future translational research to clarify the biological interaction between sarcopenia and regulation of cancer immunity.

## Supporting information

**S1 Fig. ROC curve for predicting recurrence using sarcopenia.** ROC, receiver operating characteristic.
(TIF)

**S2 Fig. OS and DFS after surgery compared between sarcopenia patients and non-sarcopenia patients in the validation cohort.** DFS, disease-free survival; OS, overall survival.
(TIF)

**S3 Fig. CIR rate after surgery between sarcopenia patients and non-sarcopenia patients in the validation cohort.** CIR, cumulative incidence of recurrence.
(TIF)

**S1 Dataset. Data set used in this study.**
(XLSX)

## Author Contributions

**Conceptualization:** Yo Kawaguchi.

**Data curation:** Yo Kawaguchi.

**Formal analysis:** Yo Kawaguchi.

**Funding acquisition:** Yo Kawaguchi.

**Investigation:** Yo Kawaguchi.

**Methodology:** Yo Kawaguchi.

**Project administration:** Yo Kawaguchi.

**Resources:** Yo Kawaguchi.

**Software:** Yo Kawaguchi.

**Supervision:** Jun Hanaoka, Yasuhiko Ohshio, Keigo Okamoto, Ryosuke Kaku, Kazuki Hayashi, Takuya Shiratori, Akira Akazawa.

**Validation:** Yo Kawaguchi.

**Visualization:** Yo Kawaguchi.

**Writing – original draft:** Yo Kawaguchi.

**Writing – review & editing:** Yo Kawaguchi.

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
