## [Decision Letter · Decision Letter 0]

7 Jun 2021

PONE-D-21-05704

Sarcopenia increases the risk of post-operative recurrence in patients with non-small cell lung cancer

PLOS ONE

Dear Dr. Kawaguchi,

Thank you for submitting your manuscript to PLOS ONE. After careful consideration, we feel that it has merit but does not fully meet PLOS ONE’s publication criteria as it currently stands. Therefore, we invite you to submit a revised version of the manuscript that addresses the points raised during the review process.

Specifically:

  Please elaborate on comments made by reviewer # 1.  Please make sure to address all  **confounding** factors that may affect the conclusion.

We look forward to receiving your revised manuscript.

Kind regards,

Amir Radfar, MD,MPH,MSc,DHSc

Academic Editor

PLOS ONE

Journal Requirements:

2. In the ethics statement in the manuscript and in the online submission form, please provide additional information about the patient records/samples used in your retrospective study, including: a) whether all data were fully anonymized before you accessed them; b) the date range (month and year) during which patients' medical records/samples were accessed.

4.Thank you for stating the following financial disclosure:

Reviewers' comments:

Reviewer's Responses to Questions

**Comments to the Author**

1. Is the manuscript technically sound, and do the data support the conclusions?

Reviewer #1: No

Reviewer #2: Yes

Reviewer #3: Yes

2. Has the statistical analysis been performed appropriately and rigorously? 

Reviewer #1: Yes

Reviewer #2: Yes

Reviewer #3: Yes

3. Have the authors made all data underlying the findings in their manuscript fully available?

Reviewer #1: No

Reviewer #2: Yes

Reviewer #3: Yes

4. Is the manuscript presented in an intelligible fashion and written in standard English?

Reviewer #1: Yes

Reviewer #2: Yes

Reviewer #3: Yes

5. Review Comments to the Author

Reviewer #1: Correlation does not necessarily imply causation, and P-value does not take this confounding into account.

In this study, the relationship between sarcopenia and the prognosis of patients with NSCLC seems to be associative, not causative. The authors failed to demonstrate the causation.

Besides, the adjustment for the important confounders is ignored, for example, adjuvant treatments (that are usually indicated for patients with Stage 2-3 NSCLC [according to the ESMO and NCCN guidelines]), smoking (that increase the rate of recurrence [DOI: 10.1183/13993003.congress-2016.PA4339] and impairs the survival [DOI: 10.1186/s12885-020-07358-3], tumor size, and nodal size [DOI: 10.1016/j.radonc.2020.07.030]. In addition, the investigators have not entered the age -as a confounding factor- to the multivariate analysis, while the mean age of patients with sarcopenia was significantly higher than others.

It seems that the investigators have tried to demonstrate the causative inference between sarcopenia and the prognosis in NSCLC -exclusively- using mathematical models. However, the causative connection is rather loose.

Reviewer #2: (I) Summary

The authors of this manuscript look into the role of sarcopenia and post-operative reccurence in NSCLC.

In a retrospective series of 256 treated patients with lobectomy and LN dissection at a single institution during a 5y span they explore the prognostic role of preoperative sarcopenia. Using a Cox`s model they provide they provide evidence of patients being at higher risk for recurrence if having sarcopenia preoperatively (HR 2.38, p=.001).

They acknowledge limitations of their work (retrospective study, no standard definition for sarcopenia) and propose future research to clarify effect of sarcopenia in cancer.

(II) Discussion of specific areas of improvement

1.) Abstract

2.) Introduction,

- line 49: reference?

3.) Materials and methods

- line 83: was any software used to assess muscle area?

- line 87: cut-off for sarcopenia was derived on the same population as it was tested on for recurrence. This is problematic and should be pointed-out in the discussion section! Additional analysis with cut-off values from standardised groups (normal population) could mitigate this problem.

4.) Results

5.) Discussion

- line 209: no set standard cut-off values for sarcopenia should be added as important limitation as other cut-offs could dramatically change results of this study!

(III) Other comments

- The discussion section, in general, could be a little more structured, but all in all a nice article to read.

- clear and informative tables and charts.

Reviewer #3: Dear Authors:

Thank you for submitting your interesting work. I do highlight the potential impact for interventional strategies in order to cope with sarcopenia in NSCLC patients such as exercise, which has shown important recurrence risk reductions in other cancer sites at different disease stages.

I would suggest to take precautions on assigning causality relationship between sarcopenia and post-operative recurrence risk in NSCLC patients taken from an observational retrospective study. You have made efforts to handle confounding risk but this should be explained and controlled thoroughly in the multivariate analysis and the discussion section in order to appropriately assign causality conclusions. Please consider revising Causality criteria available at ESMO Handbook of Interpreting Oncological Study Publications for this matter.

6. PLOS authors have the option to publish the peer review history of their article (what does this mean?). If published, this will include your full peer review and any attached files.

Reviewer #1: **Yes: **Farzad Taghizadeh-Hesary

Reviewer #2: No

Reviewer #3: **Yes: **Andrés M Acevedo

---

## [Author Response · Author response to Decision Letter 0]

6 Jul 2021

Responses to the comments from the Editor and reviewers

Editor

We would like to thank the Editor for the constructive critique that helped us improve the manuscript. We have made every effort to address the issues raised and to respond to all the comments. The text revised in response to the comments has been indicated by red font in the revised manuscript. Our detailed, point-by-point responses to the comments are provided below.

Please elaborate on comments made by reviewer # 1.

Response: We have revised our manuscript as suggested by reviewer # 1, and have responded to each of the comments individually.

Please make sure to address all confounding factors that may affect the conclusion.

 Response: As suggested, we have addressed all confounding factors that could have affected the conclusion in both, Table 2 (univariate analysis) and Table 3 (multivariate analysis).

Journal Requirements:

 Response: As suggested, we have revised our manuscript based on the style requirements of PLOS ONE.

2. In the ethics statement in the manuscript and in the online submission form, please provide additional information about the patient records/samples used in your retrospective study, including: a) whether all data were fully anonymized before you accessed them; b) the date range (month and year) during which patients' medical records/samples were accessed.

Response: As suggested, we added the following information in the Methods section and in the online submission form:

 “We accessed patients’ medical records between April 2020 and June 2021. These records were all anonymized during data collection; we used the anonymized data for our analysis.”

Response: The corresponding author has an ORCID iDs, but the following alert appears on entering the iD and password: “An identical, validated ORCID already exists in the database.” 

4.Thank you for stating the following financial disclosure:

a. Please clarify the sources of funding (financial or material support) for your study. List the grants or organizations that supported your study, including funding received from your institution.

d. If you did not receive any funding for this study, please state: “The authors received no specific funding for this work.”

 Response: The authors received no specific funding for this work. This has been specified in the cover letter as suggested.

5. Review Comments to the Author

The authors would like to thank the reviewers for the constructive critique, which has helped to improve the manuscript. We have made every effort to address the issues raised and to respond to all the comments. The text revised in response to the comments has been indicated by red font in the revised manuscript. Our detailed, point-by-point responses to the comments are provided below.

Reviewer #1: Correlation does not necessarily imply causation, and P-value does not take this confounding into account.

In this study, the relationship between sarcopenia and the prognosis of patients with NSCLC seems to be associative, not causative. The authors failed to demonstrate the causation.

Besides, the adjustment for the important confounders is ignored, for example, adjuvant treatments (that are usually indicated for patients with Stage 2-3 NSCLC [according to the ESMO and NCCN guidelines]), smoking (that increase the rate of recurrence [DOI: 10.1183/13993003.congress-2016.PA4339] and impairs the survival [DOI: 10.1186/s12885-020-07358-3], tumor size, and nodal size [DOI: 10.1016/j.radonc.2020.07.030]. In addition, the investigators have not entered the age -as a confounding factor- to the multivariate analysis, while the mean age of patients with sarcopenia was significantly higher than others.

Response: We sincerely appreciate your observations and would like to thank you suggesting the relevant studies.

Based on your observations, we have added these confounders, namely, age, performance status, smoking history, adjuvant treatment, tumor size, and lymph node size to Tables 2 and 3. As a result, the HR or P value have changed slightly; nevertheless, the independent risk factors for post-operative recurrence remained the same. We believe that adding these factors would make our results more convincing to the readers.

It seems that the investigators have tried to demonstrate the causative inference between sarcopenia and the prognosis in NSCLC -exclusively- using mathematical models. However, the causative connection is rather loose.

Response: We appreciate your observations.

We have checked the causality criteria in the ESMO Handbook, and concluded that our results only demonstrated the relationship between sarcopenia and post-operative recurrence; they did not demonstrate causation, as the causality criteria were not fulfilled in this study. 

We therefore deleted the sentence: “Sarcopenia increased the risk of post-operative recurrence in patients with NSCLC” from the conclusion, which have been revised as follows:

“Sarcopenia was the risk indicator for post-operative recurrence in patients with NSCLC. Our findings indicate the need for future translational research to clarify the biological interaction between sarcopenia and regulation of cancer immunity.”

Reviewer #2: (I) Summary

The authors of this manuscript look into the role of sarcopenia and post-operative reccurence in NSCLC.

In a retrospective series of 256 treated patients with lobectomy and LN dissection at a single institution during a 5y span they explore the prognostic role of preoperative sarcopenia. Using a Cox`s model they provide they provide evidence of patients being at higher risk for recurrence if having sarcopenia preoperatively (HR 2.38, p=.001).

They acknowledge limitations of their work (retrospective study, no standard definition for sarcopenia) and propose future research to clarify effect of sarcopenia in cancer.

(II) Discussion of specific areas of improvement

1.) Abstract

2.) Introduction,

- line 49: reference?

Response: We apologize for the error, and have added the citation accordingly (reference number [9]).

3.) Materials and methods

- line 83: was any software used to assess muscle area?

Response: We used Shade Quest View R (Fujifilm, Japan) software to assess muscle area. 

The relevant text has been revised in the Methods; the revised text reads:

“The cross-sectional areas (cm2) of the right and left psoas muscles at the L3 level were measured by manual tracing of CT data using Shade Quest View R (Fujifilm, Japan).”

- line 87: cut-off for sarcopenia was derived on the same population as it was tested on for recurrence. This is problematic and should be pointed-out in the discussion section! Additional analysis with cut-off values from standardised groups (normal population) could mitigate this problem.

Response: We sincerely appreciate your observations and suggestion. 

As suggested, we performed additional analysis using cut-off values from standardized groups, in which patients received surgery in 2016 and were observed up to 2021 (5 years); we believe that this will make our original cut-off values more plausible.

We added the following text to the Methods:

“In addition, we created another group which included patients who underwent curative lobectomy and lymph node dissection for NSCLC at our institution in 2016; this was the validation cohort. In this cohort, we analyzed the association between sarcopenia and post-operative recurrence using our PMI cut off values, and assessed the plausibility.”

We have also added the following text to the Results:

“To assess the plausibility of our original PMI cut off values, we analyzed prognosis using the validation cohort, which included 36 sarcopenia and 46 non-sarcopenia patients. The 5-year OS and DFS after surgery were 77.9% and 50.2%, respectively, in patients with sarcopenia versus 88.0% and 66.5%, respectively, in patients without sarcopenia (S2 Fig). The 5-year CIR rate was 48.0% in patients with sarcopenia versus 29.3% in patients without sarcopenia (S3 Fig). Based on these results, we considered our PMI cut off values to be plausible, as we could demonstrate the association between sarcopenia and post-operative recurrence in the validation cohort.”

This is problematic and should be pointed-out in the discussion section.

Response: As correctly observed, the lack of set standard cut-off values for sarcopenia was an important limitation.

We have therefore added the following text to the Discussion:

“Furthermore, we used original cut-off values to define sarcopenia. Therefore, the findings may not be generalized to other cut-off values or cohorts. To address this issue, we analyzed the prognosis using our cut-off values in the validation cohort; this may be considered plausible.”

4.) Results

5.) Discussion

- line 209: no set standard cut-off values for sarcopenia should be added as important limitation as other cut-offs could dramatically change results of this study!

Response: As correctly observed, the lack of set standard cut-off values for sarcopenia was an important limitation.

We have therefore added the following text to the Discussion:

“Furthermore, we used original cut-off values to define sarcopenia. Therefore, the findings may not be generalized to other cut-off values or cohorts. To address this issue, we analyzed the prognosis using our cut-off values in the validation cohort; this may be considered plausible.”

(III) Other comments

- The discussion section, in general, could be a little more structured, but all in all a nice article to read.

Response: We appreciate your observations, and have accordingly restructured the Discussion section; the text has been divided into to 5 paragraphs instead of 4, and the first sentences of each paragraph have been revised to improve clarity. 

- clear and informative tables and charts.

Response: We appreciate your careful appraisal. 

Reviewer #3: Dear Authors:

Thank you for submitting your interesting work. I do highlight the potential impact for interventional strategies in order to cope with sarcopenia in NSCLC patients such as exercise, which has shown important recurrence risk reductions in other cancer sites at different disease stages.

I would suggest to take precautions on assigning causality relationship between sarcopenia and post-operative recurrence risk in NSCLC patients taken from an observational retrospective study. You have made efforts to handle confounding risk but this should be explained and controlled thoroughly in the multivariate analysis and the discussion section in order to appropriately assign causality conclusions. Please consider revising Causality criteria available at ESMO Handbook of Interpreting Oncological Study Publications for this matter.

Response: We appreciate your observations.

We have checked the causality criteria in the ESMO Handbook, and concluded that our results only demonstrated the relationship between sarcopenia and post-operative recurrence; they did not demonstrate causation, as the causality criteria were not fulfilled in this study. 

We therefore deleted the sentence: “Sarcopenia increased the risk of post-operative recurrence in patients with NSCLC” from the conclusion, which have been revised as follows:

“In this study, sarcopenia was a risk indicator for post-operative recurrence in patients with NSCLC. Our findings indicate the need for future translational research to clarify the biological interaction between sarcopenia and regulation of cancer immunity.”

---

## [Decision Letter · Decision Letter 1]

27 Jul 2021

PONE-D-21-05704R1

Sarcopenia increases the risk of post-operative recurrence in patients with non-small cell lung cancer

PLOS ONE

Dear Dr. Kawaguchi,

Thank you for submitting your manuscript to PLOS ONE. After careful consideration, we feel that it has merit but does not fully meet PLOS ONE’s publication criteria as it currently stands. Therefore, we invite you to submit a revised version of the manuscript that addresses the points raised during the review process.

Specifically, please address comments made by reviewer #3

We look forward to receiving your revised manuscript.

Kind regards,

Amir Radfar, MD,MPH,MSc,DHSc

Academic Editor

PLOS ONE

Journal Requirements:

Additional Editor Comments (if provided):

Reviewers' comments:

Reviewer's Responses to Questions

**Comments to the Author**

1. If the authors have adequately addressed your comments raised in a previous round of review and you feel that this manuscript is now acceptable for publication, you may indicate that here to bypass the “Comments to the Author” section, enter your conflict of interest statement in the “Confidential to Editor” section, and submit your "Accept" recommendation.

Reviewer #1: All comments have been addressed

Reviewer #2: All comments have been addressed

Reviewer #3: (No Response)

2. Is the manuscript technically sound, and do the data support the conclusions?

Reviewer #1: Yes

Reviewer #2: Yes

Reviewer #3: Yes

3. Has the statistical analysis been performed appropriately and rigorously? 

Reviewer #1: Yes

Reviewer #2: Yes

Reviewer #3: Yes

4. Have the authors made all data underlying the findings in their manuscript fully available?

Reviewer #1: No

Reviewer #2: Yes

Reviewer #3: Yes

5. Is the manuscript presented in an intelligible fashion and written in standard English?

Reviewer #1: Yes

Reviewer #2: Yes

Reviewer #3: Yes

6. Review Comments to the Author

Reviewer #1: In this retrospective study, authors have delineated the association between sarcopenia and risk of post-operative recurrence in patients with non-small cell lung cancer. The authors accomplished to address all the comments.

Reviewer #2: Manuscipt “Sarcopenia increases the risk of post-operative recurrence in patients with non-small cell lung cancer”

Reviewer #2, 2nd review

The authors have corrected issues addressed in my initial review as well as issues pointed out by other reviewers.

To make the connection between sarcopenia and recurrence more plausible they have added a validation cohort from patients treated at the same institution but at a different time point. While having a validation cohort is an excellent idea, more could be gained if the cohort would be more independent (other institution, region, country…), but this would make the project much bigger.

They acknowledge the limitations of their study in the discussion section, which is now clearly structured, and provide a revised conclusion taking into account the issue of correlation / causation.

At this point I have no other issues / suggestions regarding this paper.

Reviewer #3: Dear Authors:

Thank you for providing a new version of your manuscript. I have noticed that you addressed major confounding bias by statistical means. This clarifies the interpretation that the reader can have based on your results.

I have the following minor concerns to be addressed in the final manuscript if it is to be published:

Introduction, line 50: If the statement refers to any evidence published or any own work, please provide reference accordingly.

As PLOSOne is not a Clinical cancer exclusive journal, please consider explaining disease burden that makes your research relevant.

Methods, lines 101 onwards: please indicate primary variable for statistical analysis according to the objective of the study (line 56) to prevent spurious P-value interpretation.

Results: Table 1. Please provide characterization of variables to be included in the multivariate analysis, especially Psoas muscle mass index as independent risk factor.

Discussion: Exercise is a potential intervention that could affect muscle mass, but it is not solely as should be part of multimodal intervention strategies such as nutritional assessment and treatment. Please discuss if other potential confounders increasing sarcopenia and recurrence risk relationship were not included in your study.

7. PLOS authors have the option to publish the peer review history of their article (what does this mean?). If published, this will include your full peer review and any attached files.

Reviewer #1: **Yes: **Farzad Taghizadeh-Hesary

Reviewer #2: No

Reviewer #3: **Yes: **Andres M. Acevedo

---

## [Author Response · Author response to Decision Letter 1]

10 Aug 2021

Responses to the comments from the Editor and reviewers

Journal Requirements:

Response: Thank you for this comment. We have reviewed the reference list and ensured that it is complete. None of the papers cited in our article have been retracted.

Reviewer #1: In this retrospective study, authors have delineated the association between sarcopenia and risk of post-operative recurrence in patients with non-small cell lung cancer. The authors accomplished to address all the comments.

Response: We would like to thank Reviewer #1 for their valuable advice that has helped us improve our manuscript.

Reviewer #2: Manuscipt “Sarcopenia increases the risk of post-operative recurrence in patients with non-small cell lung cancer”

The authors have corrected issues addressed in my initial review as well as issues pointed out by other reviewers.

To make the connection between sarcopenia and recurrence more plausible they have added a validation cohort from patients treated at the same institution but at a different time point. While having a validation cohort is an excellent idea, more could be gained if the cohort would be more independent (other institution, region, country…), but this would make the project much bigger.

They acknowledge the limitations of their study in the discussion section, which is now clearly structured, and provide a revised conclusion taking into account the issue of correlation / causation.

At this point I have no other issues / suggestions regarding this paper.

Response: We would like to thank Reviewer #2 for their pertinent advice that has helped to improve our manuscript.

Reviewer #3: Dear Authors:

Thank you for providing a new version of your manuscript. I have noticed that you addressed major confounding bias by statistical means. This clarifies the interpretation that the reader can have based on your results.

Response: We would like to thank Reviewer #3 for the constructive critique, which has helped improve the manuscript.

I have the following minor concerns to be addressed in the final manuscript if it is to be published:

Response: The revised text in response to the comments from Reviewer #3 is indicated in blue font in the revised manuscript. Our detailed, point-by-point responses to the comments are provided below.

Introduction, line 50: If the statement refers to any evidence published or any own work, please provide reference accordingly. 

Response: Thank you for this comment. As suggested, we have included our previously published work under reference [9].

[10] Kawaguchi Y, Hanaoka J, Ohshio Y, Okamoto K, Kaku R, Hayashi K, et al. Does sarcopenia affect postoperative short- and long-term outcomes in patients with lung cancer?-a systematic review and meta-analysis. J Thorac Dis. 2021;13(3):1358-69. doi: 10.21037/jtd-20-3072. PubMed PMID: 33841929; PubMed Central PMCID: PMCPMC8024851. As PLOSOne is not a Clinical cancer exclusive journal, please consider explaining disease burden that makes your research relevant.

Response: Thank you for your advice. We have added some general information on NSCLC and cancer-related sarcopenia to the Introduction.

Page 3, lines 42-45:

“Non-small cell lung cancer (NSCLC) is a major cause of cancer-related deaths globally [1]. Although tumor characteristics, such as histology and TNM stage, and treatment factors, such as surgery and chemotherapy, have traditionally been considered important to determine cancer prognosis, recently, patient factors like sarcopenia have started gaining attention [2]. ”

Methods, lines 101 onwards: please indicate primary variable for statistical analysis according to the objective of the study (line 56) to prevent spurious P-value interpretation.

Response: Thank you for this comment. Accordingly, we have specified that CIR was the primary variable in the Methods section:

Page 6, lines 108-109:

“CIR was defined as the interval between the date of surgery and the date of cancer recurrence (confirmed using an imaging test) and was set as the primary variable.”

Results: Table 1. Please provide characterization of variables to be included in the multivariate analysis, especially Psoas muscle mass index as independent risk factor.

Response: Thank you for this suggestion. We have now provided all variables used in the multivariate analysis, including psoas mass index, in Table 1. Furthermore, following additional analysis, we found that NSCLC patients with sarcopenia can experience more malignant disease than those without sarcopenia. We have added this information to the Discussion of the revised manuscript.

Page 15, lines 202-203:

“In the present study, NSCLC patients with sarcopenia presented with larger tumor size, larger lymph node size, and more vascular invasion than those without sarcopenia.”

Discussion: Exercise is a potential intervention that could affect muscle mass, but it is not solely as should be part of multimodal intervention strategies such as nutritional assessment and treatment. Please discuss if other potential confounders increasing sarcopenia and recurrence risk relationship were not included in your study.

 Response: Thank you for this pertinent comment. Reviewer #1 also mentioned that other potential confounders might increase sarcopenia and cancer recurrence. We have therefore added this information to the section on limitations in the Discussion of the revised manuscript.

Page 17, lines 230-233:

“Second, sarcopenia also occurs in patients suffering from malnutrition, congestive heart failure, chronic obstructive pulmonary disease, chronic renal failure, etc. [23]. Therefore, we cannot exclude the possibility that such potential confounders could have also increased sarcopenia in our patients and influenced cancer recurrence.”

---

## [Decision Letter · Decision Letter 2]

7 Sep 2021

Sarcopenia increases the risk of post-operative recurrence in patients with non-small cell lung cancer

PONE-D-21-05704R2

Dear Dr. Kawaguchi,

We’re pleased to inform you that your manuscript has been judged scientifically suitable for publication and will be formally accepted for publication once it meets all outstanding technical requirements.

Kind regards,

Amir Radfar, MD,MPH,MSc,DHSc

Academic Editor

PLOS ONE

Additional Editor Comments (optional):

Reviewers' comments:

Reviewer's Responses to Questions

**Comments to the Author**

1. If the authors have adequately addressed your comments raised in a previous round of review and you feel that this manuscript is now acceptable for publication, you may indicate that here to bypass the “Comments to the Author” section, enter your conflict of interest statement in the “Confidential to Editor” section, and submit your "Accept" recommendation.

Reviewer #1: All comments have been addressed

Reviewer #3: All comments have been addressed

2. Is the manuscript technically sound, and do the data support the conclusions?

Reviewer #1: Partly

Reviewer #3: Yes

3. Has the statistical analysis been performed appropriately and rigorously? 

Reviewer #1: Yes

Reviewer #3: Yes

4. Have the authors made all data underlying the findings in their manuscript fully available?

Reviewer #1: Yes

Reviewer #3: Yes

5. Is the manuscript presented in an intelligible fashion and written in standard English?

Reviewer #1: Yes

Reviewer #3: Yes

6. Review Comments to the Author

Reviewer #1: The authors have addressed all the comments.

There is no additional comment.

Reviewer #3: Thank you for updating your manuscript. I have no further comments or revisions to make at this point.

7. PLOS authors have the option to publish the peer review history of their article (what does this mean?). If published, this will include your full peer review and any attached files.

Reviewer #1: No

Reviewer #3: **Yes: **Andrés M Acevedo M

---

## [Editor Report · Acceptance letter]

14 Sep 2021

PONE-D-21-05704R2 

Sarcopenia increases the risk of post-operative recurrence in patients with non-small cell lung cancer 

Dear Dr. Kawaguchi:

I'm pleased to inform you that your manuscript has been deemed suitable for publication in PLOS ONE. Congratulations! Your manuscript is now with our production department. 

Kind regards, 

on behalf of

Dr. Amir Radfar 

Academic Editor

PLOS ONE